# Sepsis research in Canada: An environmental scan of sepsis investigators, research, and funding

Muhadisa Ali[1], Saad Y. Salim[2], Fatima Sheikh[3], Alison E. Fox-Robichaud[2,3,4]*,
On behalf of Sepsis Canada¶

**1** Faculty of Health Science, McMaster University, Hamilton, Ontario, Canada, **2** Department of Medicine, McMaster University, Hamilton, Ontario, Canada, **3** Department of Health Research Methods, Evidence and Impact, McMaster University, Hamilton, Ontario, Canada, **4** Hamilton Health Sciences, Hamilton, Ontario, Canada

¶ Membership of the Sepsis Canada research network can be found online at https://www.sepsiscanada.ca/.
* afoxrob@mcmaster.ca

## Abstract

Sepsis is the world's second leading cause of mortality. In 2017, the World Health Assembly declared sepsis a global priority and adopted a resolution prompting member states to improve the prevention, recognition, and management of sepsis. This cross-sectional study examines the sepsis research landscape in Canada, including demographics, scope, and funding. Using convenient sampling, sepsis researchers in Canada were asked to complete an online 20-question survey. We also scanned the CIHR funding database from 2012–2022 to quantify national research dollars spent on sepsis-related projects. Quantitative data was summarized using descriptive statistics, and textual descriptions of current sepsis research activities were analyzed thematically. With a response rate of 46% (69 of the 150), respondents were primarily men (n = 46/69, 67%), who identified as White/European (n = 49/69, 71%), and were professors or clinical professors (n = 36/69, 52%). The predominant areas of research focus were identification of sepsis (n = 21/55, 38%) and treatment/management (29/55, 53%) of sepsis, while sepsis prevention (n = 4/55, 7%) and sepsis education (n = 5/55, 9%) garnered less attention. Past 10 years of CIHR funding data revealed that only 0.7% ($85 million) of total funding ($11 billion) was towards sepsis research, of which only 2 were new-investigator awards. This study illustrates the need for improving the diversity of sepsis researchers in Canada; expanding the scope of research to address sepsis prevention, recovery, and education; and increasing overall funding to sepsis.

**Data availability statement:** Data for this survey will be available from the Open Science Framework (OSF) at the following DOI: https://doi.org/10.17605/OSF.IO/ZQG8M.

**Funding:** The authors received no specific funding for this work. Sepsis Canada is supported by the Canadian Institutes of Health Research (Funding Reference Number SRN168062).

**Competing interests:** I have read the journal's policy, and the authors of this manuscript have the following competing interests: AFR is the Scientific Director of Sepsis Canada. SS is the Managing Director of Sepsis Canada, and his salary is supported by the CIHR grant through McMaster University. The other authors have no competing interests to declare. These competing interests do not alter the authors' adherence to PLOS Global Public Health policies.

## Author summary

Sepsis is the body's extreme response to infection and a leading cause of death worldwide and in Canada. In 2017, the World Health Assembly declared sepsis a global health priority that requires action on all fronts, including by enhancing sepsis research capacity. To translate sepsis research into public health practice in Canada, we need a clear understanding of the national landscape of research. Thus, our study focused on understanding the scope of sepsis research in Canada. We surveyed 155 Canadian sepsis researchers, collecting data on their research focus and demographic diversity. We also assessed the amount of investment in Canadian sepsis research. Our findings highlight the need to: a) expand research in sepsis prevention, recovery, and education, b) improve diversity amongst Canadian sepsis researchers, and c) increase funding for sepsis research. Our findings are a call to action for key stakeholders to address these gaps, to ultimately reduce the immense burden of sepsis.

## Introduction

Sepsis, a life-threatening condition resulting from the body's extreme response to infection, is a critical global health issue. It is the world's second leading cause of mortality [1], with an estimated 49 million cases and 11 million deaths, accounting for 20% of all global deaths [2]. In Canada, sepsis is responsible for approximately 1 in 18 annual deaths [3] and incurs annual healthcare costs exceeding $2.6 billion [4].

Sepsis disproportionately affects populations with inequities in access and quality of care, and Canada is no exception. The complexity of sepsis makes it imperative to adopt a comprehensive public health approach that addresses the entire continuum of the disease, from prevention to recovery [5,6]. Accordingly, in 2017, the World Health Assembly (WHA) declared sepsis a global health priority and adopted a resolution urging member states towards an integrated approach to "improve the prevention, diagnosis, and clinical management of sepsis" [7,8]. Strategies to meet resolution WHA70.7 included public health measures such as increasing the awareness of the burden of sepsis, implementing standards of sepsis care, and enhancing sepsis research capacity.

Despite its significant impact, there is limited information on the burden of sepsis in Canada. To address these gaps, four institutes of the Canadian Institutes of Health Research (CIHR) funded Sepsis Canada – a multidisciplinary, national research network with the mandate of reducing the burden of sepsis [9]. Since 2019, Sepsis Canada has been directing efforts to provide a collaborative research platform, with a focus on increasing patient and family partner involvement, raising sepsis awareness, training the next generation of scientists, and embedding equity, diversity, and inclusion as core values driving research innovation in public health.

Research must encompass the full spectrum of sepsis to ensure public health efforts can move into practice [10]. Coordinating efforts to achieve this requires a

comprehensive understanding of the national landscape. There is currently no consolidated understanding of the trends in Canadian sepsis research, funding, or demographic characteristics of researchers. A thorough examination of these areas is essential to appropriately identify disparities in research capacity, optimize future areas of investment, and strengthen our national response towards reducing the burden of sepsis.

The objectives of this study were to understand the scope of current sepsis research conducted by Canadian researchers, including the objectives and study populations. To this end, we a) identified and described sepsis research foci across Canada, and b) evaluated the demographic diversity of sepsis researchers. A key aim of this initiative has been to assess the extent of investment in sepsis research across Canada, ensuring that future efforts are strategically aligned to improve outcomes.

## Methods

### Ethics statement

Ethical approval for this study was waived by the Hamilton Integrated Research Ethics Board. Participants received a recruitment email outlining the study's purpose, aims, and the voluntary nature of participation. Consent was implied upon survey completion and the Hamilton Integrated Research Ethics Board waived the need for formal consent.

In this cross-sectional study, a survey aimed at sepsis researchers across Canada was conducted. Collected data were securely stored online using McMaster University's secure Office 365 platform. To ensure confidentiality, all responses were de-identified and reported in aggregate form. Reporting of survey methods and results followed the Checklist for Reporting Results of Internet E-Surveys (CHERRIES) [11].

### Participants

To identify a population of independent sepsis researchers, we conducted a PubMed search using the keywords terms "sepsis" and "Canada" with a date range of 2003–2023, yielding 4000+ articles. From these articles, we extracted the author lists and cross-referenced them using institutional websites, continuing until no new authors emerged, to identify individuals with a faculty position at a Canadian institution. Inclusion criteria were authors who had contributed to sepsis-related publication and were affiliated with a Canadian institution. Graduate students, trainees, post-doctoral fellows, research staff, and patient/family partners involved in sepsis research were excluded from the study.

### Survey development

The initial set of survey questions was developed based on the study objectives and categorized into two areas: researcher demographics and research focus. Questions were then revised to formulate a 20-question survey. The final questionnaire contained two sections consisting of multiple choice and interval dropdown questions (see S1 Text for the Full Questionnaire). The first section collected demographic details while the second section asked about participants' sepsis research, including years of research involvement, publications, and research team. The questionnaire ended with an open-ended question asking participants to briefly describe their current sepsis-related research projects. Information on the size and composition of the research team, and research focus was only collected from respondents that indicated current involvement in sepsis research. Otherwise, the survey ended at number of publications for respondents who previously conducted sepsis research but had no current involvement.

The questionnaire was pilot tested by three members of the Sepsis Canada Network who were also eligible participants for the study. This involved sensibility testing to check for semantic comprehension, validity of survey questions, flow of questions, and the time required to complete the survey. Feedback from the pilot test was used to make minor language changes to the final survey. As these changes did not affect survey structure, results from the pilot test were included in the final dataset.

## Survey administration

Recruitment emails were sent to a list of eligible researchers as identified above. The survey was also distributed to knowledge users within the network, including the Canadian Critical Care Trials Group and the Canadian Critical Care Translational Biology Group. To ensure maximum outreach, the survey was shared on the Sepsis Canada X (formerly Twitter) account and through the Sepsis Canada newsletter. Participant eligibility criteria were specified in all advertisements and in the survey description. Multiple responses from the same individual, based on identical responses, were reviewed and the most recent and complete response was included. We were unable to determine whether responses originated from email solicitation or social media dissemination, as the same survey link was used across all platforms.

The survey remained open for seven weeks. Follow-up reminders were sent at the one-, three-, and six-week mark to researchers who had received the initial recruitment email and had not yet completed the survey.

Non-response bias was limited by designing a short survey with an average response time of approximately four minutes; resending recruitment emails with updated contact information in response to bounced emails; and following up with pilot testers to ensure emails were successfully received [12].

## Extraction of data from CIHR funding database

To determine the amount of research dollars spent on sepsis research, we used the CIHR's funding database to collect data on all funded awards from 2012 to 2022 [13]. Using "sepsis" as the search keyword, a comprehensive dataset was downloaded as an Excel file that included the following parameters: funding amounts, funding categories, geographical distribution, and research institutions.

## Data analysis

Descriptive statistics (frequencies, percent, medians) were used to summarize quantitative data. Specifically, the distributions of survey participants' race, gender, academic position, research team composition, years of research involvement, number of publications, and institutional affiliation were reported as frequencies and percents, by province. The median, minimum, and maximum values were calculated for the data on years of sepsis research involvement, number of sepsis-related publications, research team size, and team composition using the median for grouped data formula, as all values were intervals [14]. Primary research focus was analyzed through visualization using bar graphs.

Participant descriptions of their current sepsis-related projects were analyzed thematically by categorizing each textual response into one of the four CIHR pillars: biomedical, clinical, health systems services, and population health, based on CIHR definitions [15]. Importantly, biomedical included research up to human testing without diagnostic or therapeutic purposes. If it had a diagnostic or therapeutic intent, it was categorized as clinical. Similarly, health systems services research included factors affecting any aspect of healthcare, whereas population health research involved understanding population-based factors affecting health status.

## Results

### Response rate

Using our inclusion criteria, we were able to successfully send recruitment emails on April 5, 2023, to 150 Canadian sepsis researchers, and had the survey open until May 29, 2023. There were 73 responses to the survey (Fig 1). After removing 4 duplicates/blanks, we had a total of 69 valid responses. Despite using social media and knowledge user groups to boost our response rate, we did not see an increase in the number of respondents. As such, the denominator used to calculate response rate was the 150 respondents. This resulted in a response rate of 46%.

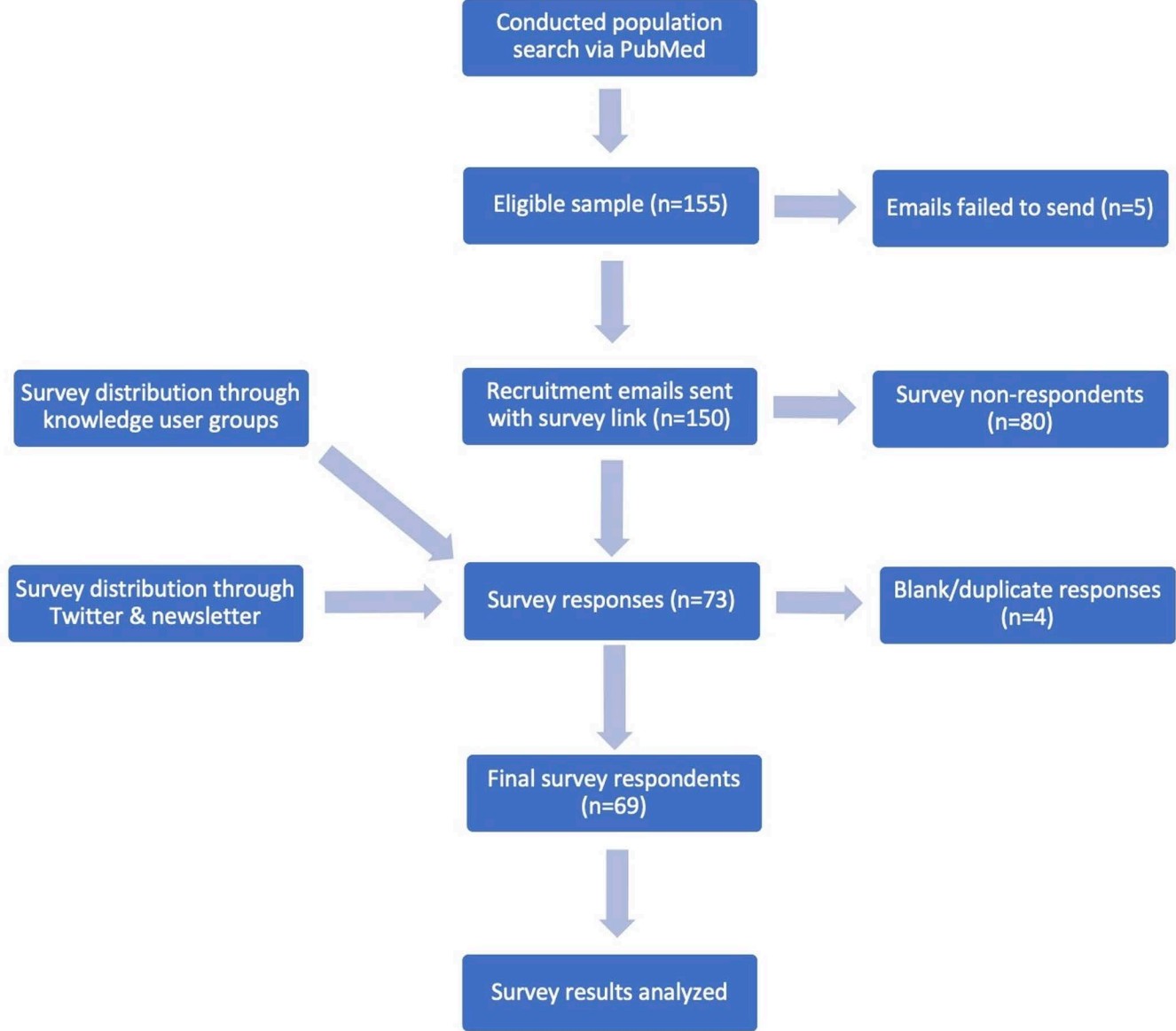

**Fig 1. Consort diagram illustrating the systematic recruitment procedure of survey participants.** The initial pool of eligible Canadian sepsis researchers was 155. After eliminating duplicates (n = 1), blank responses (n = 2), and ineligible participants (n = 1), the final cohort subjected to analysis comprised 69 respondents, yielding a response rate of 46%.

## Participant demographics

Responses were collected from 6 of Canada's 10 provinces. The highest proportion of respondents were from Ontario (n = 29/69, 42%), followed by British Columbia (n = 13/69, 19%) (Fig 2). There were no respondents from Saskatchewan, the Maritimes (besides Nova Scotia), or any of the territories.

Respondents' characteristics are presented in Table 1. Sixty-seven percent of participants identified as men and 33% as women. As shown in Fig 2, there were more men than women respondents, especially in British Columbia (11:2), Alberta (7:3), Manitoba (4:0), and Quebec (7:1). Most respondents self-identified as White/European (n = 49/69, 71%),

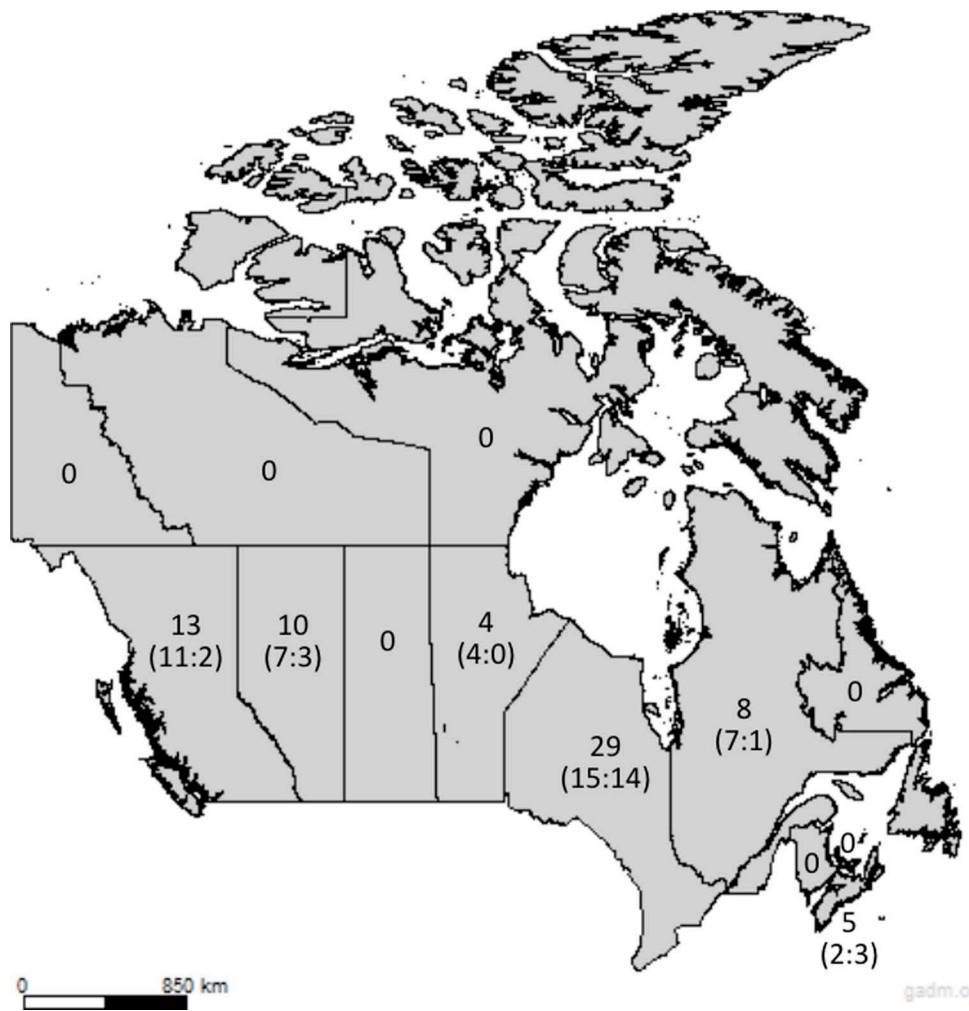

**Fig 2. Provincial Distribution and Gender Identification (Men: Women) of Respondents.** Base map source: GADM (https://gadm.org), used under the CC BY-NC 4.0 license (https://gadm.org/license.html).

followed by East Asian (n = 8/69, 12%) (Table 1). All respondents in Manitoba (n = 4/4, 100%), and Quebec (n = 8/8, 100%), identified as White/European. We had no respondents self-identified as First Nations, Inuit/Inuk, and/or Métis (0%).

Ontario had the highest number of institutions, with respondents from nine different institutes, as seen in Table 1. The greatest number of respondents from a single institution were from University of British Columbia (n = 12/69, 17%), followed by McMaster University (n = 9/69, 13%), University of Calgary (n = 8/69, 12%), and University of Toronto (n = 7/69, 10%).

The most common academic positions in both Ontario and Alberta were (senior) Professors, with Ontario also having one Professor Emeritus (Table 1). One respondent from Ontario was involved in sepsis research within Canada but held an academic position in Europe. The number of senior academic positions outnumbered the junior positions across all responding provinces. No respondents with a clinical academic position were noted from Alberta, Manitoba, or Nova Scotia.

## Research involvement and research team composition

Data on average years of sepsis research involvement, number of sepsis-related publications, research team size, and composition of Canadian sepsis researchers by province are displayed in Table 1.

**Table 1. Demographics, academic position and research metrics of respondents by province.**

| Province | Alberta | British Columbia | Manitoba | Nova Scotia | Ontario | Quebec |
|---|---|---|---|---|---|---|
| **Gender, n (%)** | | | | | | |
| Men | 7 (70) | 11 (85) | 4 (100) | 2 (40) | 15 (52) | 7 (88) |
| Women | 3 (30) | 2 (15) | 0 (0) | 3 (60) | 14 (48) | 1 (12) |
| TOTAL = 69(100) | 10(14) | 13(19) | 4(6) | 5(7) | 29(42) | 8(12) |
| **Race, n (%)** | | | | | | |
| Black/Caribbean/African | 0 | 1 (8) | 0 | 0 | 0 | 1 (13) |
| East Asian | 3 (30) | 2 (15) | 0 | 1 (20) | 2 (7) | 0 |
| South Asian | 0 | 1 (8) | 0 | 0 | 5 (17) | 0 |
| Southeast Asian | 0 | 0 | 0 | 0 | 0 | 0 |
| Hispanic/Latin American | 0 | 0 | 0 | 0 | 1 (3) | 1 (13) |
| Middle Eastern/West Asian | 0 | 0 | 0 | 1 (20) | 0 | 0 |
| White/European | 6 (60) | 9 (69) | 4 (100) | 3 (60) | 19 (66) | 8 (100) |
| Prefer not to answer | 1 (10) | 0 | 0 | 0 | 2 (7) | 0 |
| **Total Number of Institutions** | 2 | 2 | 1 | 1 | 9 | 3 |
| **Academic Positions, n (%)** | | | | | | |
| Professor* | 4 (40) | 8 (61) | 2 (50) | 1 (20) | 13 (48) | 4 (50) |
| Associate Prof. | 4 (40) | 1 (8) | 1 (25) | 4 (80) | 8 (29) | 1 (12) |
| Assistant Prof. | 2 (20) | 1 (8) | 1 (25) | 0 | 3 (11) | 0 |
| Clin. Prof. | 0 | 2 (15) | 0 | 0 | 0 | 2 (25) |
| Clin. Associate Prof. | 0 | 0 | 0 | 0 | 1 (4) | 1 (12) |
| Clin. Assistant Prof. | 0 | 1 (8) | 0 | 0 | 0 | 0 |
| Clin. Lecturer/Scholar | 0 | 0 | 0 | 0 | 1 (4) | 0 |
| Other** | 0 | 0 | 0 | 0 | 1 (4) | 0 |
| **Years of sepsis research, median (min, max)** | 8 (0, 20+) | 15 (0, 20+) | 8 (0, 20) | 8 (0, 20+) | 13 (0, 20+) | 11 (5, 20+) |
| **Sepsis-related publications, median (min, max)** | 8 (0, 25+) | 13 (0, 25+) | 8 (0, 25+) | 8 (0, 25+) | 9 (0, 25+) | 11 (0, 25+) |
| **Research team size, median (min, max)** | 7 (1, 10) | 16 (1, 20+) | 13 (1, 20) | 7 (1, 10) | 6 (1, 20+) | 4 (1, 10) |
| **Team composition, median (min, max)** | | | | | | |
| Post doc/grad trainees | 3 (0, 10) | 3 (0, 10) | 3 (1, 4) | 3 (0, 10) | 3 (0, 10) | 3 (1, 41,4) |
| Undergrad trainees | 2 (0, 5) | 2 (0, 5+) | 2 (1, 2) | 1 (0, 2) | 2 (0, 5+) | 2 (0, 5) |
| Patient partners | 0 (0, 5) | 2 (0, 5+) | 2 (1, 5) | 1 (0, 5+) | 2 (0, 5+) | 1 (0, 2) |

\* Includes 1 Professor Emeritus from Ontario.

\*\* Includes 1 researcher from Ontario who held an academic position outside of Canada.

Note: Ontario had 2 blank responses for academic position.

Approximately 80% (n = 55/69) of respondents were actively involved in sepsis research at the time of the survey. British Columbia demonstrated substantial sepsis research capacity, having the highest median number of years of sepsis research involvement (15 years), most sepsis-related publications in the past ten years (13), and largest research team size (16).

In terms of research team composition, the numbers of undergraduate, graduate, and post-doctoral trainees were relatively consistent across the country (Table 1). Patient partner involvement was in the range of 1–2 per research team for most provinces. Twenty out of the 55 respondents (36%) reported 0 patient partners as part of their research team. With a median of 0, Alberta had the least patient partner involvement in their research teams.

 

PLOS Global Public Health

## Primary research focus

Sepsis treatment/management (n = 29/55, 53%), followed by identification of sepsis (n = 21/55, 38%), were the most popular primary research foci among respondents (Fig 3). Conversely, sepsis prevention (n = 4/55, 7%) and sepsis education (n = 5/55, 9%) garnered less attention. When examining the demographics/groups targeted by the respondents in their studies, we observed that most researchers focused on individuals over 65 years (n = 35/55, 64%) and the impact of biological sex (n = 24/55, 44%) (Fig 4). In contrast, gender-related sepsis effects (n = 6/55, 11%) and research on Equity, Diversity, Inclusion, Decolonization (EDID) (n = 11/55, 20%) received comparatively less research consideration.

In analyzing respondents' descriptions of current sepsis-related research projects through the lens of the CIHR pillars, we found a range of research foci within each pillar (Table 2). Predominant topics included animal models and neonatal sepsis (biomedical), pediatric sepsis and COVID-19 mechanisms of sepsis (clinical), sepsis awareness and barriers to care (health systems services), and sepsis epidemiology (population health). A detailed analysis is available in S1 Data.

## Funding allocation to sepsis research

An analysis of the CIHR funding database from 2012 to 2022 (10 years) revealed a total allocation of $85,126,000 for sepsis-related research (see Table 3). The majority of funds, $56,200,000, supported investigator-led projects in the form

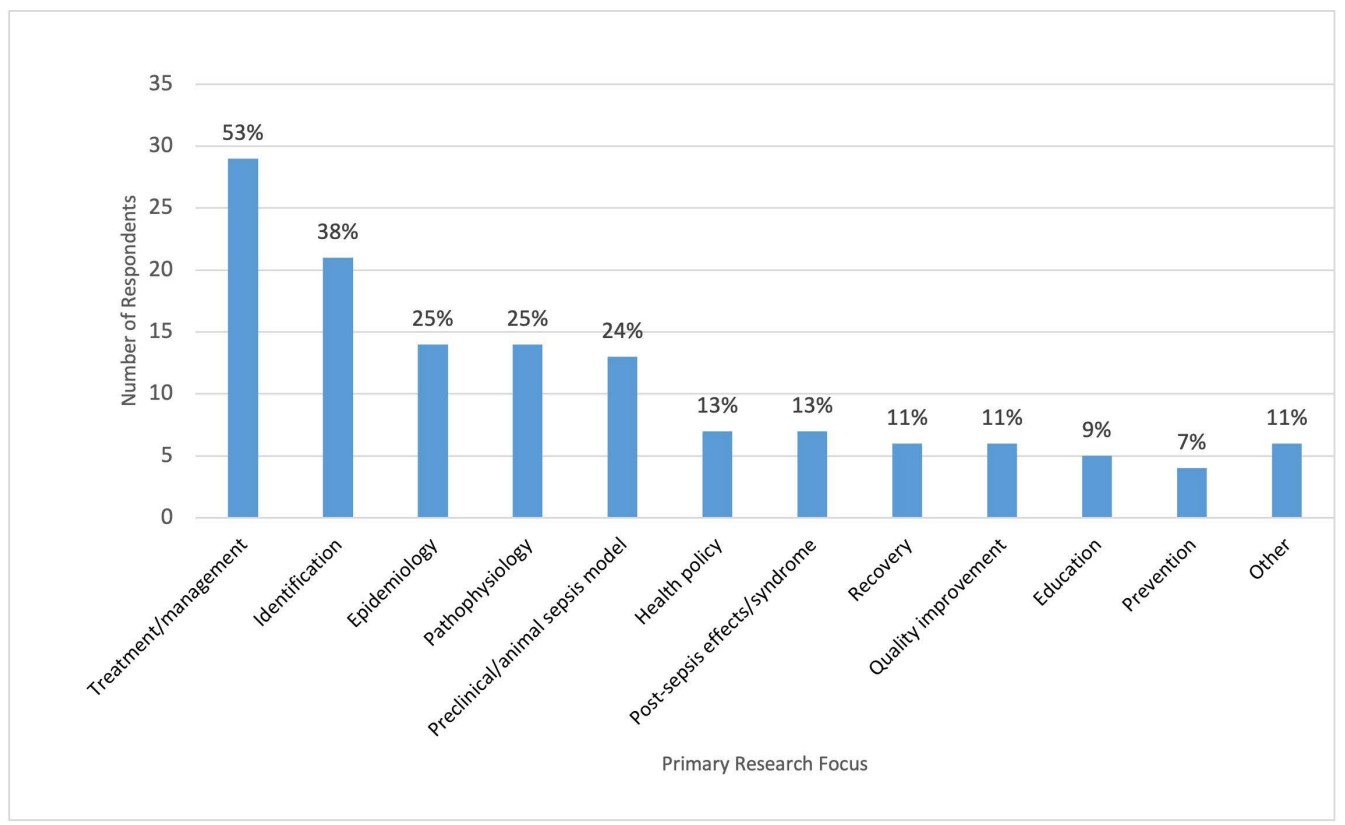

**Fig 3. Primary research focus of respondents.** The multi-select survey question "What is your primary research focus? Please select all that apply", was posed only to participants who had active research program/project in sepsis.

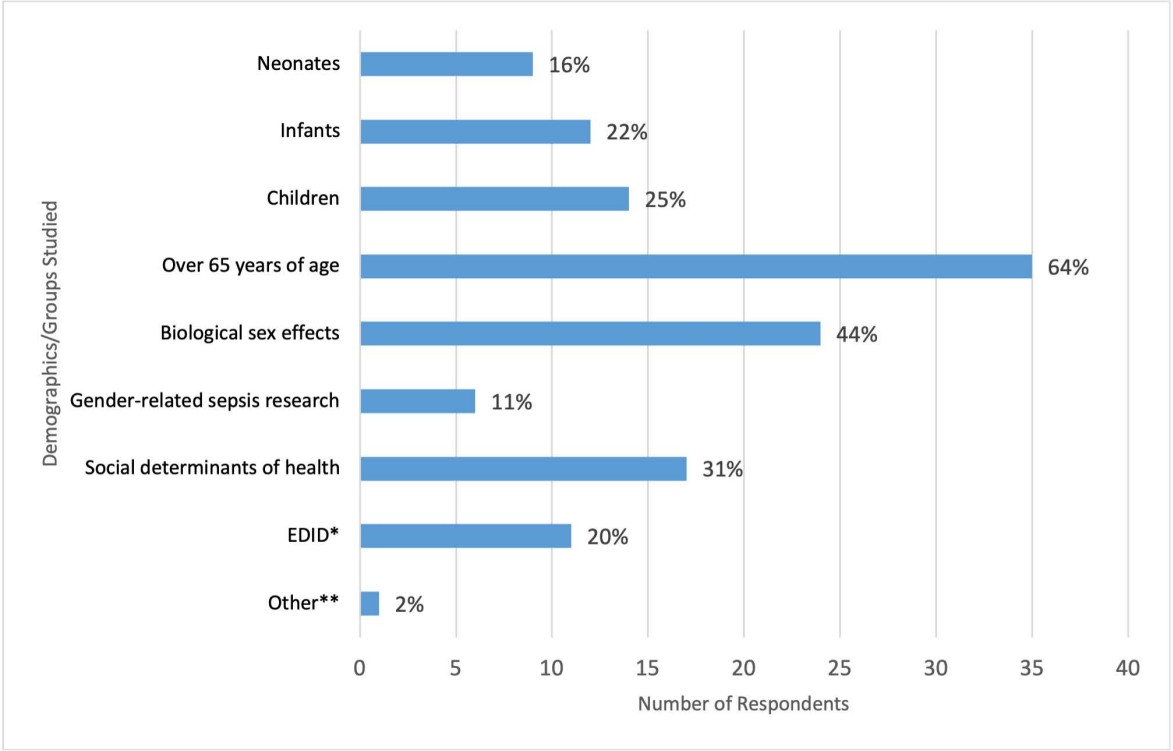

**Fig 4. Demographics/groups investigated by respondents.** *EDID = Equity, Diversity, Inclusion, Decolonization. **Other = Sepsis murine models.

**Table 2. Primary research focus of Canadian sepsis researchers categorized by CIHR pillars via thematic analysis of respondent descriptions of current research projects.**

| Biomedical | Clinical | Health Systems Services | Population Health |
|---|---|---|---|
| Animal Models | Pediatric sepsis | Sepsis awareness | Neonatal sepsis epidemiology |
| Acute Lung Injury | COVID-19 mechanisms & impacts of sepsis | Training | Social determinants in pediatric sepsis |
| Neutrophil, macrophage pathways | Improving antibiotic use | Barriers to care | Provincial outcomes |
| Bloodstream infection/ bacteremia | Fluid therapy | Health care resource utilization | |
| Cannabinoids | Hemodynamic monitoring | Quality improvement | |
| Respiratory infection (pneumonia, viral) | Respiratory infection (pneumonia, viral) | | |
| Acute kidney injury | Acute kidney injury | | |
| Neonatal sepsis | Neonatal sepsis | | |

of operating grants, including COVID-19-related projects that totalled $7,507,000 between 2020 and 2021. Other notable observations include 2 new investigator awards and 54 trainee awards (including scholarships, travel grants, and fellowships), totaling $4,126,000. Note that Canada had several competitions exclusively for COVID-19 research, including COVID-19 Rapid Research Funding Competitions and these funding allocations were not included in the calculations [16].

**Table 3. CIHR Funding of Sepsis Research between 2012-2022. Using CIHR's funding database, the keyword "sepsis" was used to download all awardees. The table below shows the category, number of awards and total amounts awarded by CIHR. Total CIHR funding in those years was $10,851,000,000.**

| Category | Number | Amount |
|---|---|---|
| Trainee awards (scholarships, fellowships) | 54 | $ 4,126,000.00 |
| Operating Grants (incl. COVID) | 79 | $ 56,200,000.00 |
| Foundation Grants | 7 | $ 21,200,000.00 |
| New Investigator Grants | 2 | $ 300,000.00 |
| CHRP | 4 | $ 1,300,000.00 |
| Others (dissemination, catalyst) | 19 | $ 2,000,000.00 |
| **TOTAL** | **165** | **$ 85,126,000.00** |
| Team Grant (Sepsis Canada) | 1 | $ 5,700,000.00 |

## Discussion

Sepsis is a significant global public health concern and WHA's adoption of a resolution to 'improve the prevention, diagnosis, and clinical management of sepsis', has prompted member states to take proactive measures in funding sepsis-related research initiatives. Simultaneously, in an era marked by the necessity for prudent public spending and judicious allocation of resources, understanding the landscape of sepsis research is important. Funded by CIHR, Sepsis Canada was mandated to unify experts from various disciplines into an integrated and cohesive research program that spans the four CIHR pillars. The findings of this study serve as a tool to discern the nature, extent, and gaps in sepsis research in Canada, and may serve as a valuable reference for identifying similar trends, challenges, and opportunities in other jurisdictions.

Despite an increase in the number of researchers interested in sepsis research, there was a lack of consolidated understanding of the trends in sepsis research, funding, and diversity of researchers. Of the survey respondents, 50% of the researchers focused primarily on identifying and treatment/management of sepsis, particularly in individuals over the age of 65 years. This finding aligns with sepsis-associated mortality rates in Canada (between 2009 and 2011), as 28.9% of all deaths were among individuals over the age of 65 [3]. Conversely, 28.7% of sepsis-associated deaths were in children under 15 years old, and 35% of respondents focused their research in this age category.

Our analysis revealed disparities in the distribution of sepsis researchers across provinces, with no respondents from the territories. This has potential implications for equitable funding allocations. However, the importance of demographic diversity extends beyond mere representation; it encompasses the broader issue of addressing disparities in healthcare outcomes for different racial and ethnic groups, particularly visible minorities, and Indigenous peoples. As highlighted by Hennessy et al. [17], visible minorities in Canada, including Indigenous populations, are at increased risk of sepsis. This underscores the importance of having researchers from diverse backgrounds, as they may bring unique perspectives and insights to addressing these disparities [18]. The lack of diversity in the field may stem from differential attainment or "opportunity gap" seen in medical education and training. Menezes et al. [19] highlights the challenges in academic and career achievement experienced by specific racial groups, often referred to as racial disparities in attainment. This concept is crucial to understanding the challenges faced by researchers from underrepresented backgrounds in the field of sepsis research.

In addition to the diversity issue and the disparities in sepsis research across the country, there is a concern regarding the pipeline of junior sepsis researchers. This is exemplified by the fact that CIHR has invested in only two new investigator grants focused on sepsis in the past decade, highlighting the need for more support and mentorship for this critical field. Sepsis Canada has attempted to address this gap by investing significant portion of its funds in developing and

expanding its training program in collaboration with the Life-Threatening Illness National Group (LifTING). This program trains multidisciplinary researchers, patient and family partners, and community members in research methods, professional skills, and attitudes related to sepsis and life-threatening illnesses.

Compounding the disparities in researcher diversity, a concerning dearth of funding for sepsis research is evident. In the Canadian context, the allocation of funds for sepsis research by CIHR between 2012 and 2022 amounted to $85 million, which, when compared to the total funding disbursed during the same period at $11 billion [20], represents only 0.7% of total funding. Compared to the United States, the National Institutes of Health (NIH) awarded 1.7% of its total ($29.7 billion) funding to sepsis research, between 1997–2012 (noting that these fund allocations were prior to any COVID funding) [21,22]. These figures underscore a notable underinvestment in sepsis research, despite the significant burden of sepsis on healthcare systems. It is possible that the consequences of this underinvestment are apparent in Ontario, where health care costs related to sepsis were estimated at $1 billion per year prior to the pandemic [4]. In fact, the British Columbia Sepsis Network demonstrated that investments in sepsis awareness and care can yield a return of $112.50 for every dollar spent, and more importantly, they saved over 150 lives from sepsis-associated deaths [23].

Our study has several limitations. First, there is the potential for self-selection bias, as participation was voluntary, which may result in the inclusion of individuals with stronger motivations or experiences in sepsis research. Consequently, the findings may not fully represent the entire spectrum of sepsis researchers across Canada. Furthermore, reliance on self-reported data introduces the possibility that respondents may not have disclosed all their research activities comprehensively. Secondly, being a cross-sectional study, it provides a snapshot of the current state of sepsis research in Canada. It does not allow for the analysis of trends over time or provide insights into the longitudinal development of the field. Understanding such temporal variations, particularly considering potential pre- and post-pandemic impacts on sepsis research, is an area where our study has limited applicability and is important to consider for next steps.

In conclusion, the global health impact of sepsis and the World Health Organization's call for intensified research efforts highlight the urgent need to address this critical issue. Our study provides valuable insights into the current landscape of sepsis research in Canada, emphasizing the necessity to bridge knowledge gaps and support initiatives that extend beyond biomedical and clinical studies. Our findings underscore the importance of fostering an inclusive research environment that prioritizes diversity, particularly in addressing disparities in healthcare outcomes for various racial and ethnic groups. Equally crucial is the allocation of adequate resources to understand, prevent, and manage sepsis effectively.

Our study serves as a call to action for key stakeholders to address disparities, embrace diversity, and substantially increase investments in sepsis research. The insights gained from the Canadian context can inform and inspire similar efforts in other jurisdictions, ultimately contributing to a global strategy aimed at reducing the burden of sepsis.

## Supporting information

**S1 Text. Full questionnaire.**
(DOCX)

**S1 Data. Thematic analysis of research focus: CIHR pillars.**
(DOCX)

## Author contributions

**Conceptualization:** Alison Fox-Robichaud.

**Data curation:** Muhadisa Ali, Saad Y. Salim.

**Formal analysis:** Muhadisa Ali, Saad Y. Salim.

**Methodology:** Muhadisa Ali.

**Supervision:** Fatima Sheikh, Alison Fox-Robichaud.

**Writing – original draft:** Muhadisa Ali, Saad Y. Salim.

**Writing – review & editing:** Muhadisa Ali, Saad Y. Salim, Fatima Sheikh, Alison Fox-Robichaud.

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
