## [Decision Letter · Decision Letter 0]

11 Feb 2025

PGPH-D-24-01704

Sepsis Research in Canada: An Environmental Scan of Sepsis Investigators, Research, and Funding

Dear Dr. Fox-Robichaud,

Thank you for submitting your manuscript to PLOS Global Public Health. After careful consideration, we feel that it has merit but does not fully meet PLOS Global Public Health’s publication criteria as it currently stands. Therefore, we invite you to submit a revised version of the manuscript that addresses the points raised during the review process. Specifically the reviewer has requested a number of clarifying questions that should be addressed.

We look forward to receiving your revised manuscript.

Kind regards,

Amy Huei-Yi Lee

Academic Editor

Journal Requirements:

1. Please provide additional details regarding participant consent. In the ethics statement in the Methods and online submission information, please ensure that you have specified (1) whether consent was informed and (2) what type you obtained (for instance, written or verbal, and if verbal, how it was documented and witnessed). If your study included minors, state whether you obtained consent from parents or guardians. If the need for consent was waived by the ethics committee, please include this information.

2.  Please provide an Author Summary. This should appear in your manuscript between the Abstract (if applicable) and the Introduction, and should be 150–200 words long. The aim should be to make your findings accessible to a wide audience that includes both scientists and non-scientists. Sample summaries can be found on our website under Submission Guidelines:

https://journals.plos.org/globalpublichealth/s/submission-guidelines#loc-parts-of-a-submission.

3. Figure 2: please (a) provide a direct link to the base layer of the map (i.e., the country or region border shape) and ensure this is also included in the figure legend; and (b) provide a link to the terms of use / license information for the base layer image or shapefile. We cannot publish proprietary or copyrighted maps (e.g. Google Maps, Mapquest) and the terms of use for your map base layer must be compatible with our CC-BY 4.0 license. 

Additional Editor Comments (if provided):

Reviewers' comments:

Reviewer's Responses to Questions

**Comments to the Author**

1. Does this manuscript meet PLOS Global Public Health’s publication criteria ? Is the manuscript technically sound, and do the data support the conclusions? The manuscript must describe methodologically and ethically rigorous research with conclusions that are appropriately drawn based on the data presented.

Reviewer #1: Yes

2. Has the statistical analysis been performed appropriately and rigorously?

Reviewer #1: Yes

3. Have the authors made all data underlying the findings in their manuscript fully available (please refer to the Data Availability Statement at the start of the manuscript PDF file)?

Reviewer #1: No

4. Is the manuscript presented in an intelligible fashion and written in standard English?

Reviewer #1: Yes

5. Review Comments to the Author

Reviewer #1: This is an interesting study that surveyed Canadian sepsis non-trainee researchers regarding their demographics and research foci. This study also investigated investment in sepsis research by CIHR in Canada. Overall, this study identified a lack of diversity in our sepsis researchers, limited research in the area of sepsis prevention/recovery education, and an under investment in sepsis research in Canada.

INTRODUCTION:

Line 44 and line 46: indicate that these are annual (?) estimates for case and death counts and healthcare costs.

Line 48-51, up to and including "....2017.": The Canada specific information seems out of place in this paragraph. I would move this info into the subsequent paragraph where you talk about why Sepsis Canada was formed. The line about CMAJ can probably be deleted entirely.

Last paragraph of the Introduction should be edited to include that one aim was assessing extent of investment in sepsis research in Canada.

METHODS

Is there a chance that there are eligible participants identified via dissemination through twitter, the CCCTG, and the CCCTBG, who would NOT have been captured via the PubMed search that was done to look for participants? If so, this needs to be clarified under Participants.

How did you go about determining who were faculty vs. not in the author lists for articles found by your PubMed search? This seems like a lot of work.

Under "Survey Development" - Briefly describe details around i) demographics, and ii) sepsis research (beyond pubs, research team, years active).

Provide the link to the CIHR funding database if it is publicly available.

RESULTS

The Methods indicate that eligible participants were in part identified via a PubMed search. How many articles did you get with your search to be able to identify the 155 people you sent the survey to to? Do you have a sense of gender/academic positions of these eligible folks, and how those who didn't respond differ in terms of these characteristics compared to those who did respond?

Did all participants answer all questions? Or were there blanks?

Line 183/184: include numerators and denominators, not just percentages.

Table 1: footnote indicates that one participants had a cross appointment in Portugal - risks identifying participant, could just say "abroad" as opposed to Portugal.

Line 240-247 incl Table 2: How many participants fall into each CIHR pillar?

Table 1: Add a column for "Total" across all provinces.

Figure 3: This figure seems unnecessary - all this info is in Table 1. Would remove.

Figure 4: I would order this by frequency. What is the difference between post-sepsis syndrome and recovery - could these categories be collapsed? Include # (%) over the bars.

Figure 5: Needs a footnote with acronym meanings (EDID?) and also what "Other" includes. Include # (%) over the bars.

DISCUSSION:

Line 275: I am unsure what this sentence means?

Line 284: I would also mention that nobody seems to be working on this in the territories.

I think it is worth discussion that most respondents are senior faculty - why might this be? Should we worry about our pipeline of junior sepsis researchers?

Consider discussing strategies to improve DEI in sepsis research.

6. PLOS authors have the option to publish the peer review history of their article (what does this mean? ). If published, this will include your full peer review and any attached files.

**Do you want your identity to be public for this peer review?** For information about this choice, including consent withdrawal, please see our Privacy Policy .

Reviewer #1: No

---

## [Editor Report · Decision Letter 1]

31 Mar 2025

Sepsis Research in Canada: An Environmental Scan of Sepsis Investigators, Research, and Funding

PGPH-D-24-01704R1

Dear Dr. Fox-Robichaud,

We are pleased to inform you that your manuscript 'Sepsis Research in Canada: An Environmental Scan of Sepsis Investigators, Research, and Funding' has been provisionally accepted for publication in PLOS Global Public Health.

Best regards,

Amy Huei-Yi Lee

Academic Editor